# Molecular Mechanisms Driven by MT4-MMP in Cancer Progression

**DOI:** 10.3390/ijms24129944

**Published:** 2023-06-09

**Authors:** Emma Muñoz-Sáez, Natalia Moracho, Ana I. R. Learte, Alice Collignon, Alicia G. Arroyo, Agnés Noel, Nor Eddine Sounni, Cristina Sánchez-Camacho

**Affiliations:** 1Department of Health Science, School of Biomedical Sciences, Universidad Europea de Madrid, Villaviciosa de Odón, 28670 Madrid, Spain; emma.munoz@universidadeuropea.es; 2Department of Medicine, School of Biomedical Sciences, Universidad Europea de Madrid, Villaviciosa de Odón, 28670 Madrid, Spain; natalia.moracho@universidadeuropea.es; 3Department of Dentistry, School of Biomedical Sciences, Universidad Europea de Madrid, Villaviciosa de Odón, 28670 Madrid, Spain; anaisabel.rodriguez@universidadeuropea.es; 4Laboratory of Biology of Tumor and Developmental Biology, GIGA Cancer, Liège University, B-4000 Liège, Belgiumagnes.noel@uliege.be (A.N.); 5Cancer Metabolism and Tumor Microenvironment Group, GIGA Cancer, Liège University, B-4000 Liège, Belgium; 6Department of Molecular Biomedicine, Centro de Investigaciones Biológicas Margarita Salas (CIB-CSIC), 28040 Madrid, Spain; agarroyo@cib.csic.es; 7Department of Genetics, Physiology and Microbiology, Faculty of Biological Sciences, Complutense University of Madrid, 28040 Madrid, Spain

**Keywords:** MT4-MMP, MMP-17, development, vasculature, tumor, cancer cell, malignancy, therapy, biomarker

## Abstract

MT4-MMP (or MMP-17) belongs to the membrane-type matrix metalloproteinases (MT-MMPs), a distinct subset of the MMP family that is anchored to the cell surface, in this case by a glycosylphosphatidylinositol (GPI) motif. Its expression in a variety of cancers is well documented. However, the molecular mechanisms by which MT4-MMP contributes to tumor development need further investigation. In this review, we aim to summarize the contribution of MT4-MMP in tumorigenesis, focusing on the molecular mechanisms triggered by the enzyme in tumor cell migration, invasiveness, and proliferation, in the tumor vasculature and microenvironment, as well as during metastasis. In particular, we highlight the putative substrates processed and signaling cascades activated by MT4-MMP that may underlie these malignancy processes and compare this with what is known about its role during embryonic development. Finally, MT4-MMP is a relevant biomarker of malignancy that can be used for monitoring cancer progression in patients as well as a potential target for future therapeutic drug development.

## 1. Introduction

Matrix metalloproteinases (MMPs) are a family of zinc-dependent endopeptidases that are able to degrade and remodel the extracellular matrix (ECM) by the cleavage of distinct matrix components and also to lead to the proteolytic activation/inactivation of receptors, growth factors, adhesion molecules, cytokines, and other pericellular proteins [1,2]. Included within the MMP family are the membrane-type matrix metalloproteinases (MT-MMPs) that are anchored to the cell membrane by a type I transmembrane domain (MT1-MMP/MMP-14, MT2-MMP/MMP-15, MT3-MMP/MMP-16, and MT5-MMP/MMP-24) or by a glycosylphosphatidylinositol (GPI) anchor (MT4-MMP/MMP-17 and MT6-MMP/MMP-25) [1,3,4].

Membrane-type 4 MMP (MT4-MMP, also known as MMP-17) is anchored by a GPI motif to the plasma membrane, which confers exclusive mechanisms of biosynthesis and regulation. Apart from this, MT4-MMP conserves the three structural domains characterized in all MMPs: the prodomain, the catalytic, and the hemopexin domains. The prodomain, with a length of 80 amino acids and a consensus sequence with unpaired cysteines, keeps the enzyme in a latent state, named zymogen. The catalytic domain is located at the C-terminal of the prodomain. It has a conserved sequence (“HEXXHXXGXXH”) that includes a Zn^2+^ ion binding motif which is essential for the proteolytic activity of the proteinase. This is linked by the hinge region to the hemopexin domain involved in substrate recognition and degradation. Next, MT4-MMP displays a small region called the stem region which has two cysteines. These residues appear to be involved in the dimerization and oligomerization of MT4-MMP [5] in a similar manner to that described for the other GPI-anchored metalloproteinase (MT6-MMP) [4]. Once MT4-MMP is exposed at the cell surface, its enzymatic activity is regulated by endogenous inhibitors, tissue inhibitors of metalloproteinases (TIMPs). There are four mammalian TIMPs (TIMP-1, 2, 3, and 4) that inhibit MT-MMPs by binding their N-terminal domain with the catalytic zinc ion of the enzyme. MT4-MMP is inhibited by all TIMPs, among which TIMP-1 is the most effective inhibitor [2,4,6].

Interestingly, the hemopexin domain of MT4-MMP only displays a 40% similarity with the same domain of the other family members [7]. This feature may explain the specificity and exclusivity of the substrates to be cleaved by MT4-MMP and potentially its internalization mode [1,5,8]. In fact, only a limited number of substrates have been identified for this endopeptidase, including matrix and matricellular proteins such as fibrin/fibrinogen, gelatin [8], osteopontin [9], periostin [10], proteases such as ADAMTS-4 [11,12], and membrane proteins such as αM-integrin [13], Pro-TNFα [8], and LRP [14]. In addition, the type of anchoring to the cell membrane presented by MT4-MMP confers a localization in specific lipid domains that will condition its biological activity [2].

Due to their role in ECM degradation and remodeling of the pericellular microenvironment, the MMP family of proteases contributes to several physiological processes such as embryogenesis, organogenesis, tissue regeneration, angiogenesis, and wound healing. In particular, MT4-MMP expression has been reported during angiogenesis, limb development, and in distinct brain regions from early embryonic to postnatal stages [15] as well as in adult tissues such as the brain, ovaries, testis, and colon [7,16,17,18,19]. However, its physiological role remains unclear.

The MMP family also plays a crucial role in different pathological processes such as arthritis, cardiovascular disease, and cancer progression [4]. Regarding MT4-MMP involvement in tumor development, this protease was first detected in human breast carcinoma [18] exerting pro-angiogenic and pro-metastatic functions [20,21]. In addition, MT4-MMP-mediated cancer dissemination has been described in other types of tumors including colon and head and neck squamous cell cancer (HNSCC) [22,23]. Interestingly, its role in cancer progression may be linked to MT4-MMP’s contribution to the stability and permeability of the tumor vasculature [2,4]. All this evidence opens the possibility for the use of MT4-MMP as an interesting potential therapeutic target against tumor progression.

## 2. MT4-MMP Expression and Regulation in Tumors

### 2.1. Pattern of Expression of MT4-MMP in Normal and Tumor Tissues

Most MT-MMPs play an important role in physiological conditions. Regarding MT4-MMP, although its function has been described mainly in relation to tumor cell growth [18,20,24], its expression is spatiotemporally controlled during embryonic development. In mice, MT4-MMP is expressed in a dynamic pattern of expression from early to postnatal stages of development with a high expression of this enzyme during vascular and limb development and brain formation [15]. These data suggest that this metalloproteinase may be associated with novel functions in angiogenesis, endocardial formation, and vascularization during organogenesis as well as in central nervous system (CNS) development in correlation with its expression pattern [15]. In this context, MT4-MMP is essential for the proper organization of the aortic vessel wall. Thus, MT4-MMP expression was confirmed in vascular smooth muscle cells (VSMCs) during aortic development where, subsequently, the proteinase contributes to cell differentiation and maturation through its catalytic activity [9,15] (Figure 1A).

During CNS development, the enzyme localizes to certain regions such as the olfactory bulb, cerebral cortex, and hippocampus, suggesting a key role in brain development [15,17] (Figure 1A). Interestingly, MT4-MMP knockout mice seem to show no apparent defects in gestation, growth, fertility, and behavior and showed no evident abnormal developmental phenotypes [16]. Despite this, MT4-MMP is highly expressed in the kidney papilla and the anterior hypothalamus, and null mice display a decreased intake of water and daily urine output, suggesting a role for this enzyme in water homeostasis and the regulation of the thirst center [25].

It is also worth noticing that during embryonic development, MT4-MMP is essential for neural crest cell (NCC) migration in zebrafish [26]. In the mouse embryo, MT4-MMP is also expressed in premigratory and migratory NCCs at early stages of development [15]. This seems relevant for the regulation of cell migration, as shown in zebrafish, where it is known that its orthologue mmp17b interacts with the MMP inhibitor RECK which is required for the proper NCC migration [26].

In adult human tissues, the MT4-MMP protein was expressed preferentially in different brain regions such as the cerebral cortex, the hippocampal formation, and the basal ganglia. Additionally, this metalloproteinase is detected in female tissues (including the uterus, cervix, and ovary) where it participates in endometrial angiogenesis during the menstrual cycle [27] and in the gastrointestinal tract, particularly in the colon, under physiological conditions. In fact, among the cells that express this protein are excitatory and inhibitory neurons, oligodendrocyte precursor cells, smooth muscle cells, melanocytes, and monocytes [28,29].

Most of our current information regarding MT4-MMP comes from cancer studies. Indeed, MT4-MMP was cloned for the first time from human breast carcinoma cells [18]. Thereafter, various proteomic studies have reported the expression of this proteinase in different tumors such as prostate and oral carcinomas, lung cancer, cervical carcinomas [30,31], osteosarcomas, embryonal carcinomas, leukemia [32], adrenal adenocarcinomas, and thyroid cancer [31]. In addition, melanoma accumulates a high expression of MT4-MMP possibly related to its presence in the skin and connective tissue in physiological conditions [28,29,32]. Transcriptomic data on the expression of twenty-four MMPs, including MMP-17 from the Cancer Genome Atlas (TCGA) were reported by Gobin et al., 2019 in fifteen cancer types [33]. In this study, differential gene expression of MT4-MMP was found to increase by more than two folds in head and neck cancer, renal clear cell cancer, lung adenocarcinoma, and in lung squamous cancer. In all breast cancers, the MT4-MMP transcript was found to increase by 1.34-fold with a significant p-value. While MT4-MMP (MMP-17) is not universally upregulated through all cancer types as happens in other MMPs (such as MMP-1, MMP-9, MMP-10, MMP-11, MMP-13, and MMP-14), the study of the prognostic value of 24 MMPs in predicting overall survival in the 15 cancer types has revealed MT4-MMP along with MMP-14, and MMP-23B as the most frequently MMPs exclusively associated with poor prognosis when they are overexpressed in a particular cancer [33].

In this context, MT4-MMP has been linked to cancer dissemination. For instance, in vitro studies and subcutaneous xenografts have confirmed the association between overexpression of MT4-MMP with cell proliferation [20,24]. In the same line, high levels of MT4-MMP expression in gastric tissues are associated with lymph node metastasis and serosal involvement, and therefore, with tumor invasion [34]. In lung metastasis, MT4-MMP alters blood vasculature and induces pericyte detachment, promoting tumor dissemination [21]. In contrast, MT4-MMP is downregulated in glioma development. In this sense, as the tumor grade advances, the expression of the proteinase continues decreasing [35]. MT4-MMP downregulation is unique to glioma because in other cancer cell lines (Jurkat or HeLa among others) MT4-MMP expression levels are higher. One possible explanation is that glioma cells are different from other cancer cells because they rarely metastasize. Moreover, brain invasion by glioma cells is very extensive locally with a large vascular development in which MT4-MMP could play an additional role. It might be interesting to analyze the specific mechanisms by which MT4-MMP downregulation favors glioma progression perhaps due to this dual role compared to other tumor cells. Furthermore, it is described that MT4-MMP is predominantly expressed by glioma cells, instead of microglia, which is the key driver for the tumor cell invasion in the CNS [36].

Interestingly, MT4-MMP is not only restricted to cancer cells, since positive staining was also observed in nearby cancer stromal cells, supporting its importance in the process of tumor invasion and metastasis. All this evidence opens the possibility of using MT4-MMP as an interesting potential therapeutic target against tumor progression.

### 2.2. Regulation of MT4-MMP Expression and Activity in Normal and Tumor Tissues

Although it is known that the above-mentioned MT4-MMP functions are regulated at various levels, such as gene expression, compartmentalization, pro-enzyme cleavage, and substrate processing [37,38], little is known about the detailed molecular mechanisms relevant to tissue growth and expansion. However, its proteolytic activity has been shown to be significantly linked to its pro-metastatic activity, and the latter may explain certain aspects of its role in cancer progression (Figure 1B). For example, in contrast to most of MT-MMPs, MT4-MMP hydrolyses very few ECM components [1], and it also exhibits characteristic sensitivity to TIMPS as well as inefficient activation of pro-MMP2, which may explain its singular role in promoting tumor progression.

#### 2.2.1. Transcription

Regarding gene expression, classical methods have been enriched by using a novel technology known as super-resolution microscopy (SRM) that allows the study of gene regulation in a much more detailed way [39]. The use of SRM approaches could be interesting for the study of MT4-MMP transcription mechanisms that are not yet fully understood.

It has been reported that MT4-MMP expression is induced by hypoxia through the hypoxia-inducible factor-1-α (HIF1-α) and the activation of SLUG, a known transcription factor involved in epithelial–mesenchymal transition (EMT) which promotes the malignant capacity of cancer cells (Figure 1B) [22]. In fact, invadopodia formation and amoeboid movements, which are both crucial mechanisms to promote metastatic dissemination, are mediated by HIF1-α-induced MT4-MMP expression in head and neck cancer tumor cells [40]. An increase in MT4-MMP has been also observed under hypoxic conditions or under constitutive expression of HIF1-α in other types of tumors such as hypopharyngeal squamous cell carcinoma (FADU) and tongue squamous cell carcinoma (SAS) [22]. Interestingly, SLUG has been identified as the key factor responsible for the hypoxia-induced MT4-MMP expression through the activation of the proteinase promoter by interacting with its E-box [22]. Notably, SLUG expression was mostly restricted to migrating neural crest cells and several mesodermal derivatives in the embryo [41], suggesting that this transcription factor may regulate MT4-MMP expression both during development and tumorigenesis. Therefore, co-expression of MT4-MMP and HIF-1α may be considered as an indicator of breast cancer prognosis. It is also worth noting that in human breast cancer, MT4-MMP transcription is also regulated by the methyltransferase hSED1A, which appears to be over-expressed in these circumstances. It is known that silencing hSED1A decreases MT4-MMP transcription, which impairs cell migration and invasion of tumor cells on lung tissue and colon cancer cells [42].

#### 2.2.2. Post-Translational Regulation

Compartmentation

As a GPI-anchored protein, MT4-MMP is located on lipid domains that seem to be relevant for its activity [2]. In line with this, the HM-7 cancer cell line expresses MT4-MMP in lipid rafts, while caveolin-1 is not detected. Interestingly, restoration of caveolin-1 expression in metastatic HM-7 cells inhibits MT4-MMP localization to lipid rafts, thereby suppressing the metastatic phenotype of HM-7 colon cancer cells. These findings raise the possibility that MT4-MMP compartmentation may be directly or indirectly involved in certain intracellular signaling events that control its pericellular proteolytic capacity [23].

The functional cooperation between MT4-MMP and other GPI proteins such as the urokinase receptor uPAR may be possible since both colocalize in the same microdomains [7]. Although this role during embryonic development remains elusive, this putative co-expression could be relevant for tumor invasion in several cancers [34]. Indeed, MT4-MMP requires a permissive microenvironment to exert its tumor-promoting effect. Tumor-derived MT4-MMP cannot circumvent the absence of a host angio-promoting factor such as the plasminogen activator inhibitor-1 (PAI-1), which cooperates with the uPA/uPAR axis in different contexts [43].

Internalization and recycling

Endocytosis is the mechanism that controls the amount of MT4-MMP anchored to the cell surface through the CLIC/GEEC pathway as well as recycles it back to the cell membrane [5,44]. The signaling cascade that triggers this endocytic pathway involves the Rho family GTPase and Cdc42 and the transcription factors Arf1 or GBF1 that are responsible for the regulation of Cdc42 activity. For instance, MT4-MMP internalization was shown to be primarily dependent on Cdc42 and RhoA, and to a lesser extent on Rac1 in MDA-MB-231 cells, a human breast cancer cell line overexpressing the proteinase. These data suggest that MT4-MMP proficiently uses the CLIC/GEEC pathway (rather than the caveolin-dependent or clathrin-dependent pathways) for its internalization [5]. It should be mentioned that the actin-binding protein Swiprosin-1 (Swip1) functions as a cargo-specific adaptor for CLIC/GEEC endocytic pathway mediating the endocytosis of active integrins, supporting integrin-dependent cancer cell migration and invasion [45]. As MT4-MMP can also regulate the levels and activity of integrins, particularly β2-integrins, in other cellular contexts [13], this endocytic pathway may be particularly relevant for regulating MT4-MMP activity in different cell types and physio-pathological contexts.

Dimerization

A relationship between MT-MMP dimerization and greater proteolytic activity in tumor cells is feasible [46,47]. MT4-MMP is found in homodimers and oligomers at the cell surface maintained via disulfide bond between the cysteine residues of the stem region [4,48]. In the context of tumorigenesis, MT4-MMP can form homodimers depending on Cys574 and the formation of disulfide bonds between the monomers both in transfected non-tumor MDCK and COS1 cells and in MD-MB-231 breast carcinoma cells [5,48]. Although the impairment of dimerization did not decrease cell invasion in vitro, it could still be relevant for other activities in the tumor context.

Shedding

Shedding is another alternative mechanism used by MT-MMPs to control their pericellular proteolytic activity once they are anchored to the cell membrane [3,49]. This mechanism may either involve the release of the extracellular portion of the active MT-MMP to the cellular milieu or the removal of the enzyme from the cell surface. However, the precise mechanism of MT4-MMP shedding remains to be elucidated and it is not fully understood how MT4-MMP regulates the balance between the amount of protein anchored to the membrane by its GPI moiety and the soluble enzyme since it is not affected in the presence of TIMPs [4,7]. The possibility that this metalloproteinase could be released from the cell surface through the activity of a phosphatidylinositol (PI) specific phospholipase C, similarly to other membrane dipeptidases, cannot be ruled out [7].

Interactions

EGFR has been reported to associate with MT4-MMP by co-immunoprecipitation. Both EGFR and MT4-MMP have been shown to cooperate in tumor cell invasion and signaling, driving cancer cell growth through the regulation of cell cycle proteins such as CDK4 activation and retinoblastoma protein inactivation (Figure 1B) [24]. MT4-MMP is thought to stimulate cell proliferation by interacting with EGFR and enhancing its activation by its ligands, the epidermal growth factor (EGF), and tumor growth factor (TGF) in cancer cells (Figure 1B). Whether this functional cooperation is also relevant during embryonic development, for example in the heart formation [50], remains to be investigated.

## 3. Cellular Processes and Molecular Mechanisms Induced by MT4-MMP in Cancer Progression

### 3.1. Tumor Cell-Autonomous MT4-MMP Actions

#### 3.1.1. Tumor Cell Migration and Invasiveness

Studies on mouse embryonic development have shed light on the role of the MT4-MMP gene in the physiological EMT process, in which epithelial cells lose polarity and acquire motility during brain development and heart morphogenesis [15,26,51]. EMT is a developmental process that induces important changes in cellular polarity leading to the transition of epithelial cells into mesenchymal cells [52]. MT-MMPs exert a relevant role during EMT along different species. For instance, the catalytic activity of MT1-MMP regulates cadherin levels and promotes the transition and migration of NCCs in Xenopus [53]. Similarly, the orthologous of MT4-MMP in zebrafish (mmp17b) is also required for the proper migration of NCC since embryos lacking mmp17b exhibit a defective developmental trunk patterning [26].

MT4-MMP activity is regulated within spatiotemporal frames in normal tissue, likely to maintain normal physiological function. However, its overexpression in breast cancer cells led to enhanced lung metastasis and tumor vessel destabilization in xenografts [20,21]. In line with these data, immunohistochemical studies revealed overexpression of MT4-MMP in breast adenocarcinomas [20], in triple-negative breast cancer (TNBC) [24,54], and in gastric cancer [34]. Its expression was associated with the depth of tumor invasion, lymph node metastasis, serosal involvement, and transperitoneal spread of gastric cancer cells [34]. MT4-MMP-dependent cancer cell migration and invasion in vitro has been attributed to its dimerization through cysteine residues [48]. Its specific cell surface localization in lipid rafts has been associated with a dynamic toward a migratory phenotype of colon cancer cells [31]. While MT4-MMP was inefficient in promoting breast cancer cell migration and invasion in vitro and in the 2D matrix [20], it promotes metastasis in vivo. More recently, Yan et al. reported that MT4-MMP expression in head and neck squamous cell carcinoma (HNSC) increases invadopodia formation and gelatin degradation [40]. Mechanistically, the authors show that MT4-MMP binding with Tks5 and PDGFRα results in Src activation and promotes amoeboid-like movement by stimulating the small GTPases Rho and Cdc42. Together, these data support the hypothesis that MT4-MMP induces, in a cell-dependent manner, crucial mechanisms of cancer migration and invasion, the key element of cancer metastasis. Similarly, overexpression of MT1-MMP and MMP-2 in human papillomavirus-infected cells is related to invasiveness in cervical carcinoma. In this context, it has been described that viral oncoproteins transcriptionally regulate the expression levels of these MMPs affecting cell invasion [55]. Further research and docking studies are still needed to analyze new potential substrates for MT4-MMP to help increase our understanding of the specific mechanisms by which it modifies cell–ECM interactions and controls cell adhesion and invasion.

#### 3.1.2. Tumor Cell Proliferation and Growth

Several proteinases can control the bioavailability and activity of growth factors and regulate cell proliferation. Mechanistically, MMPs release active molecules from the ECM, shed cell surface receptors, and process growth factor binding proteins through their proteolytic activity, which leads to angiogenesis and tumor growth [38,56,57]. For instance, MT1-MMP modulates tyrosine kinase receptors (TKR) such as receptors of platelet-derived growth factor (PDGFR) and fibroblast growth factor (FGFR2), while MMP-9 and MMP-2 activate transforming growth factor-beta (TGFβ) and release vascular endothelial growth factor (VEGF) sequestered into the ECM [57,58]. In contrast to the well-known MT1-MMP-mediated outside-in cell signaling, the signaling role of GPI-anchored MT4-MMPs in promoting cancer cell proliferation is unknown. A direct contribution of MT4-MMP to tumor growth has been demonstrated by analyzing the proliferation index (Ki67 positivity) of MT4-MMP xenografts compared to control tumors [24]. In this study, MT4-MMP was demonstrated to promote cancer cell proliferation only in 3D cell cultures and through the activation of the epidermal growth factor receptor (EGFR) after its binding to ligands. MT4-MMP was validated as a key signaling precursor and partner of EGFR, which enhances its activation leading to cancer cell proliferation in a non-proteolytic manner. In contrast to its intrinsic non-proteolytic effect on EGFR activation and cell proliferation in 3D Matrigel, the enzymatic activity of MT4-MMP was still required for the early angiogenic switch during breast cancer progression [43]. Breast cancer xenografts with an inert form of MT4-MMP (E249A) were not able to induce early angiogenesis and have reduced growth potential when compared to xenografts producing the active form of the enzyme. Interestingly, MT4-MMP has been recently identified as a biomarker for TNBC patient responses to chemotherapy [59] and to the combination of EGFR inhibitor, Erlotinib, and Palbociclib, an inhibitor of cyclin-dependent kinases 4 and 6, which are involved in the cell cycle [54]. These data highlight the proliferative effect of MT4-MMP that is dependent on EGFR signaling and the retinoblastoma tumor suppressor pathway (RB pathway) in TNBC (Figure 1B). Furthermore, they shed new light on the putative interest of MT4-MMP to predict the response of patients to some chemotherapies and targeted treatments.

### 3.2. Non-Tumor Cell Autonomous MT4-MMP Actions

One of the most important aspects of tumor cell resistance and recurrence is the microenvironmental changes that occur during tumorigenesis [60]. This involves the transformation of the pericellular ECM by different cell types, including fibroblasts, neuroendocrine and immune cells, as well as changes in the blood vessels and lymphatic network to orchestrate the precise conditions for tumor development [61].

#### 3.2.1. Alteration of the Vascular Architecture

Given that MT-MMPs can release growth factors sequestered in ECM or cell surface-associated growth factors, they can shape tumor blood vessels in the tumor microenvironment [38,62]. One of the most important roles of MMPs in vessel maturation and stability has been attributed to MT1-MMP in several studies reported on knockout mice and in models of tissue injury [62]. MT1-MMP regulates vessel stability through the increase in TGFb availability and signaling through ALK5 in blood vessels [62]. In tumor angiogenesis, pericytes take an important role in stabilizing the newly formed microvessels being recruited around the endothelial cell sprouts to form larger perfused microvessels [62,63]. In fact, in well-established xenografts of breast carcinoma, MT4-MMP in tumor cells promotes paracrine pericyte detachment from tumor vessels, thereby allowing cancer cell escape and intravasation into blood vessels and leading to metastatic spread [20,21,62]. Notably, MT4-MMP expressed by tumor cells was also found essential in promoting an early angiogenic switch and tumor blood vessel formation that has been shown to be dependent on its catalytic function. In Matrigel plug assays using cancer cells expressing the inert form of the enzyme, the tumor fails in recruiting new endothelial cells and inducing tumor vascularization when compared to tumors expressing the active form of the enzyme [43]. The MT4-MMP substrates for this pericyte/angiogenic effect in tumors are not yet known. Nevertheless, osteopontin proteolysis and JNK signaling that have been reported to be driven by MT4-MMP during vascular smooth muscle cell (VSCM) migration in the aorta vessel wall [9], might be relevant in the tumor vascular context (Figure 1A). MT4-MMP function in the vascular wall is evident as it is expressed in periaortic progenitors during embryogenesis, suggesting a role in the early formation of the aortic wall and VSMC maturation [9,15]. MT4-MMP knockout mice have dysfunctional VSMCs and dilated aortas [9] and displayed adventitial fibrosis and hypotension, like mouse models of thoracic aortic aneurysms and dissections (TAAD) (Figure 1A) [64]. Recently, periostin has been identified as a novel substrate for SMCs in the stem cell niche of the intestine [10] with actions in intestinal homeostasis and repair. Whether the MT4-MMP–periostin axis plays a role in tumor cell development and progression deserves further investigation. More recently, the absence of MT4-MMP has been shown to promote proliferation in VSMCs and the formation of collateral arterioles by induction of p38 MAPK signaling and mitochondria dynamics particularly in contexts of hypoxia or starvation (Figure 1A) [65]. The relevance of this pathway in tumors presenting heterogeneous hypoxia might be an interesting aspect to investigate.

#### 3.2.2. MT4-MMP and the Tumor Immune Response

A permissive host microenvironment during cancer progression is necessary not only for endothelial cells but also for the distinct immune cell types. In this sense, Host et al. demonstrated that MT4-MMP functions as a key intrinsic tumor cell determinant that contributes to the elaboration of a permissive microenvironment for metastatic dissemination [43]. MT4-MMP has been involved not only in angiogenesis but is also associated with inflammatory pathologies such as osteoarthritis and atherosclerosis. Both angiogenesis and inflammation are two crucial processes for tumor development. For instance, MT4-MMP levels are elevated inside immune system tissues such as in metastatic lymph nodes [20] and leukocytes [66].

During the inflammatory process, cytokine production by immune cells is associated with cancer in multiple ways, although information on the contribution of MT4-MMP to the tumor immune response is very limited [67]. MT4-MMP can be produced by immune cells promoting angiogenesis and retaining chemotactic molecules to attract them to the tumor neighborhood. Thus, MT4-MMP can activate proinflammatory cytokines such as TNF-α which is expressed in T-cells and macrophages. MT4-MMP-mediated cleavage of pro-TNF-α in isolated macrophages in vitro can trigger their activation [68]. On the contrary, Rikimaru et al. described that the contribution of MT4-MMP to the shedding of TNF-α appears to be negligible since its levels in the culture medium were similar in MT4-MMP-positive and negative macrophages after LPS induction [16]. Interestingly, many tumors produce abundant TNF-α that promotes cancer cell survival through the activation of the NF-κB signaling pathway [69]. Therefore, it would be interesting to evaluate the involvement of MT4-MMP in this pathway through the cleavage and availability of active TNF-α in tumoral cells.

MT4-MMP is also expressed in tumor-associated macrophages (TAMs) [60]. In peritumor and tumor samples of hepatocellular carcinoma, MT4-MMP-positive cells colocalized with macrophages expressing M2 phenotypic markers, while is absent in tumor cells, fibroblasts, or M1 macrophages [60]. In addition, MT4-MMP-positive TAMs reduced, in a paracrine manner, E-cadherin expression in favor of mesenchymal N-cadherin, and vimentin in hepatocellular carcinoma cells, which is characteristic of the epithelial–mesenchymal transition (EMT) necessary for tumor metastasis [60].

MT4-MMP has also been shown to regulate the crawling and recruitment of a subset of circulating monocytes, patrolling monocytes, to inflamed endothelial sites by cleaving the αMβ2 integrin adhesion receptor [13]. While these patrolling monocytes contribute to combat circulating tumor cells in the lung [70], they can also differentiate into pro-angiogenic tumor-associated macrophages once they transmigrate to the tissue [71]. Therefore, the possible contribution that MT4-MMP regulation of patrolling monocyte behavior may have on lung metastasis remains to be explored.

### 3.3. MT4-MMP and Tumor Metastasis

One of the critical steps in human cancer progression is the malignant conversion and the acquisition of invasive phenotype. A study conducted to determine factors of epithelial breast invasiveness has shown that MT4-MMP is expressed during the transition from preinvasive to invasive state [72]. Cancer cells isolated from human samples and grown in 3D cell cultures in vitro or in mouse xenografts showed an upregulation of MT4-MMP during the acquisition of an invasive phenotype. This supports the functional significance of this enzyme as a key element in tumor malignant conversion. Accordingly, MT4-MMP expression in TNBC cells stimulated cell growth in 3D cell culture, whereas it has no effect on cell proliferation or migration in 2D cell cultures [20,24]. In the context of cancer malignancy, activation of EMT is a critical phenotypic event that promotes metastatic behavior. As mentioned before, a link between hypoxia, EMT, and the regulation of MT4-MMP expression has been evidenced in head and neck cancer where SLUG contributes to HIF-1α dependent MT4-MMP [22] (Figure 1B). The expression of MT4-MMP and HIF-1α in these tumors was associated with poor overall survival of cancer patients [22]. Furthermore, the epigenetic regulation of MT4-MMP among other MMPs by the histone methyltransferase, hSETD1A, overexpressed in metastatic human breast cancer cell lines and patients has been demonstrated to drive breast cancer metastasis [73]. Generally, increased MT4-MMP activity is frequently detected in breast, colorectal, and head and neck cancer cells and fosters an invasive and metastatic phenotype [2,48,74].

## 4. MT4-MMP as a Biomarker and Therapeutic Target in Cancer Growth

The implication of MT4-MMP in cancer progression is related to its capacity to regulate intrinsic tumor cell proliferation and tumor blood vessel destabilization phenomenon. These two roles can potentiate its tumor and metastatic effects, making it an excellent target. Future studies aiming at targeting these two functions will further define their relevance in cancer and diseases. Nevertheless, the pro-proliferative effect of MT4-MMP on breast cancer cells in vitro has been inhibited by erlotinib, and the presence of MT4-MMP in tumors was shown to be associated with more sensitivity to chemotherapy [59]. Furthermore, MT4-MMP/EGFR/RB axis has been used to define a subtype of TNBC that can effectively respond to erlotinib and palbociclib combination therapy [54]. The clinical relevance of MT4-MMP, and its new partners EGFR and RB has been validated in human TNBC samples and in patient-derived xenografts of TNBC (PDX-TNBC) [2]. TNBC response to erlotinib and palbociclib was evaluated in vitro, in vivo, in xenografts and PDX models of TNBC and demonstrated that MT4-MMP/EGFR/RB axis is highly relevant in 50% of TNBC. PDX of this TNBC subtype showed complete inhibition of tumor growth upon combined erlotinib and Palbociclib treatment. Together, these useful data on MT4-MMP/EGFR/RB association and therapy combination yield clinically relevant insights and provide potentially useful biomarkers to identify TNBC subtypes that respond to specific therapeutic combinations. Given the high prevalence of TNBC-producing MT4-MMP and EGFR, the combination of an EGFR inhibitor with an MT4-MMP inhibitor might improve TNBC treatments. However, targeting MMPs in cancer has been proven difficult since the disappointing clinical trials reported 20 years ago [75]. The adequate task of targeting this enzyme should consider its partners such as EGFR, JNK signaling, and the molecular heterogeneity of breast cancers. In the era of personalized medicine, standardized procedures for MT4-MMP expression assessment could help in defining the right patient who could respond to anti-EGFR targeted therapy. While genomic profiling did not reveal a particular interest in MT4-MMP, immunohistochemistry studies have demonstrated its overexpression in breast cancers, suggesting possible alternative splicing in MT4-MMP transcripts in human cancers. Another consideration is to check for MT4-MMP partners that enhance cancer cell and stroma cell migration and malignancy. In this field, particular attention should be given to the generation of functional blocking molecules for MT4-MMP and exploring their combination with EGFR inhibitors in breast cancer.

## 5. Conclusions

Taken together, these data emphasize the relevance of MT4-MMP in the context of cancer. Although there are many ways by which MT4-MMP participates in tumor progression, the mechanisms used by this proteinase for epithelial–mesenchymal transition and vessel stability in tumors seem to recapitulate those described during embryonic development. However, further investigations are essential to improve our knowledge about the function of MT4-MMP in the embryo as well as in physiological and pathological conditions including cancer progression. In this regard, we want to bring particular attention to the protein expression of MT4-MMP in malignancy. The need for further investigation by bioinformatics and the use of human cancers from selected patients before and after therapy, and the association with clinicopathological features can be useful for deciphering its roles in human cancers. This will help us to develop new therapeutic approaches and to improve the design of new selective MT4-MMP inhibitors to achieve faster diagnosis and more personalized treatment for cancer patients and for vascular diseases.

## Figures and Tables

**Figure 1 ijms-24-09944-f001:**
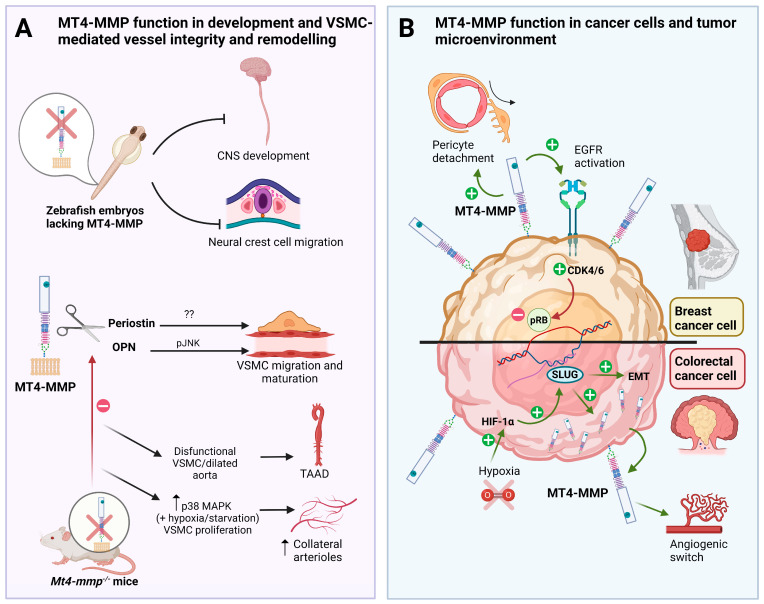
Role of MT4-MMP in embryonic development, VSMC-mediated vessel stability and cancer progression. (**A**) MT4-MMP is required for central nervous system (CNS) development and neural crest cell migration in the zebrafish embryo. MT4-MMP is expressed in VSMCs which cleaves osteopontin (OPN) to promote JNK signaling and the proper migration and maturation in the aorta. Dysfunctional VSMCs from MT4-MMP knockout mice, dilate aortas and lead to adventitial fibrosis and hypotension, playing a potential role in thoracic aortic aneurysms and dissections (TAAD). The role of periostin cleavage by MT4-MMP in VSMCs and TAAD is not known yet. Moreover, the absence of MT4-MMP promotes VSMC proliferation via p38 MAPK signaling in hypoxic/starvation context leading to an increased formation of collateral arterioles post-ligation. (**B**) MT4-MMP expression by breast cancer cells promotes EGFR signaling and pRB inactivation leading to tumor growth in triple-negative breast cancer (TNBC). MT4-MMP also promotes breast cancer metastasis by inducing pericyte detachment and vessel destabilization in a paracrine manner. In colorectal carcinoma, MT4-MMP expression is regulated by hypoxia and HIF1-α through SLUG and promotes cancer development since its expression is also required for early tumor angiogenic switch and tumor growth. Created with Biorender.com, accessed on 11 September 2022.

## Data Availability

Not applicable.

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
