# Peer review of "Molecular Mechanisms Driven by MT4-MMP in Cancer Progression"

_ijms, 2023, doi:10.3390/ijms24129944_

Round 1

Reviewer 1 Report

Molecular Mechanisms Driven by MT4-MMP in Cancer Progression

Muñoz-Sáez et al.

This manuscript is a review article discussing one of the GPI-anchored membrane-type MMPs, MT4-MMP. It covers all the findings so far made on the topics, and I enjoyed reading it.  The structure and contents are good. However, I have some problems with English. The expression, grammatical error, and style. I recommend re-checking the English, especially the expression and structures of sentences. It should be edited by a native English-speaking academic person. As it is, the article is difficult to read.

Specific points:

1.     Page 2, line 59-62. The author described: “Thus, MT4-MMP dimerization 59  through a disulphide bond via the cysteine residues of the stem region, can regulate the 60  amount of MT4-MMP at the cellular surface, as well as increase its stability and enzymatic 61  activity [3,5].” This notion is not entirely correct. It is not known if dimer formation is required for cell surface expression and enzymatic activity of MT4-MMP. The paper cited describes MT6-MMP, but not MT4-MMP. Authors need to rephrase it by describing it more accurately.

2.     Page 2, line 68. The authors described: “Interestingly, MT4-MMP has unique characteristics compared to other members of the family in terms of sequence homology, substrate specificity, and internalization mode.” I was not sure the sequence differences were “unique characteristics.”  It needs a more straightforward explanation.

3.     Page 7, line 341. “stabilizing the newly form microvessels being…” should be “stabilizing the newly formed microvessels being…”

4.     Page 7, line 354. “Thus,” should be removed. 

5.     Page 8, line 381-384. “On the contrary, Rikimaru et al., described that in Mt4-mmp-/- mice the release of TNF-α from macrophages after LPS injection was like that in wild type animals and that the expression of MT4-MMP mRNA was repressed [16].” This sentence is misleading and should be rephrased by improving English. 

These are some examples, and general English expressions need improvement throughout.

The paper has good content. However, the English style is difficult to read and sometimes confusing. 

Reviewer 2 Report

In this review, the authors have gathered interesting and novel information about the importance and contribution of MMP-17 in tumorigenesis. This review will provide info about an important MMP, which has functional roles in cancer initiation and malignant progression and could be interesting for a broad range of scientists including molecular biologists, oncologists, and cancer researchers. My major comment is that the authors could have mentioned what specific human malignancies have high expression of this important MMP, does it any correlation with the clinical feature of that malignancy, and is there any specific inhibitor for this MMP, could the author check the expression of this MMP at the mRNA and protein levels in human malignancies in the TCGA dataset, what are the most important signaling pathways that control MMP-17. 

Reviewer 3 Report

The review "Molecular Mechanisms Driven by MT4-MMP in Cancer Progression" by Muñoz-Sáez et al. focuses on the cancer-related aspects of the many roles of MT4-MMP in cellular physiology. In this specific focus, the present review is distinct from other review focused more broadly on MT4-MMP recently published in IJMS (Yip et al., 2019).

In the introduction section the authors clearly introduce the subject and propose unanswered question that they aim to answer in the following sections of the manuscript. The manuscript has only one figure, but this figure is quite complex panel and sufficiently illustrates the discussed topic.

Could the authors discuss why isMT4-MMP overexpressed in one cancer typa and downregulated in another cancer type? For instance lines 147-148 "...the association between over- 147 expression of MT4-MMP with cell proliferation." vs. line 152 "In contrast, MT4-MMP is downregulated in glioma development."

The transcription is very broad topic addressed by progressive research. Although it is difficult to introduce this topic broadly, before addressing specific aspects of MT4-MMP, the authors could use some recent reviews, such as Hoboth et al., 2021 IJMS https://doi.org/10.3390/ijms22136694 to open the topic of transcription to the broader readership.

Lines 295-297 "Further investigations are still needed to determine mechanisms driven by MT4-MMP in cell migration and in which context it can modify cell-ECM interaction and control cell adhesion and invasion." Could the authors be more specific and suggest some research direction(s) in this regard please, such as they nicely did in lines 366-367 "The relevance of this pathway in tumors presenting heterogeneous hypoxia might be an interesting aspect to investigate."?

Could the authors discuss also the link between human papillomavirus-induced cancers and MT4-MMP incl. references such as Kaewprag  et al., 2013 PLOS ONE https://doi.org/10.1371/journal.pone.0071611 or Rattay et al., 2023 J Med Virol https://doi.org/10.1002/jmv.28658 please?
